Living on the edge: genetic structure and geographic distribution in the threatened Markham’s Storm-Petrel (Hydrobates markhami)

Norambuena Heraldo V. buteonis@gmail.com 1 2 3
Rivera Reinaldo 2 4
Barros Rodrigo 3
Silva Rodrigo 3
Peredo Ronny 3
Hernández Cristián E. 2 5
1 Centro Bahía Lomas, Facultad de Ciencias, Universidad Santo Tomás , Concepción , Chile
2 Laboratorio de Ecología Evolutiva y Filoinformática, Departamento de Zoología, Facultad de Ciencias Naturales y Oceanográficas, Universidad de Concepción , Concepción , Chile
3 Red de Observadores de Aves y Vida Silvestre de Chile , Santiago , Chile
4 Millennium Institute of Oceanography (IMO), Universidad de Concepción , Concepción , Chile
5 Universidad Católica de Santa María , Arequipa , Perú
Pimm Stuart
Electronic publication date: 2021 Dec 24
Publication date: 2021
Volume: 9
Electronic Location ID: e12669
Received 2021 May 11; Accepted 2021 Dec 1
Copyright: ©2021 Norambuena et al.
Copyright year: 2021
Copyright holder: Norambuena et al.
License: This is an open access article distributed under the terms of the Creative Commons Attribution License, which permits unrestricted use, distribution, reproduction and adaptation in any medium and for any purpose provided that it is properly attributed. For attribution, the original author(s), title, publication source (PeerJ) and either DOI or URL of the article must be cited.
License URL: https://creativecommons.org/licenses/by/4.0/

Keywords: Biodiversity, Ecology, Genetic structure, Hydrobatidae, Phylogenetics, Phylogeography

Funding: American Bird Conservancy #B2016-04 #1854A) The Mohammed Bin Zayed Fund Packard Foundation FONDECYT-POSTDOCTORADO 3190618 FONDECYT 1170815 1201506 Millennium Institute of Oceanography (IMO), University of Concepción, Concepción, Chile This work was supported by the American Bird Conservancy (Grant #B2016-04 and #1854A), the Mohammed Bin Zayed Fund, Packard Foundation, FONDECYT-POSTDOCTORADO 3190618 (Heraldo V Norambuena), and FONDECYT 1170815 and 1201506 (Cristián E Hernández). Reinaldo Rivera was supported by the Millennium Institute of Oceanography (IMO), University of Concepción, Concepción, Chile. The funders had no role in study design, data collection and analysis, decision to publish, or preparation of the manuscript.

==============================
Migratory birds are threatened by habitat loss and degradation, illegal killings, ineffective conservation policies, knowledge gaps and climate change. These threats are particularly troubling in the Procellariiformes (Aves), one of the most endangered bird groups. For “storm-petrels”, their cryptic breeding behavior, asynchrony between populations, and light pollution pose additional threats that contribute to increased mortality.Markham’s Storm-Petrel (Hydrobates markhami), a poorly known migratory species, is a pelagic bird that breeds in dispersed colonies in the Sechura and Atacama Deserts, with asynchronous reproduction between colonies, and is highly affected by artificial lights. Considering its complex conservation scenario and singular breeding, we expected to find narrow habitat distribution conditions, strong geographic genetic structure, and spatially differentiation related to human population activities (e.g., light pollution) and the climate global change. To evaluate these predictions, we analyzed the phylogeography, current and future potential distribution based on mitochondrial gene ND1 and geographic records.The phylogeographic analyses revealed three well-supported clades (i.e., Paracas, Arica, and Salar Grande), and the geographical distribution modeled using an intrinsic conditional model (iCAR) suggests a positive relationship with the mean temperature of the wettest quarter and of the driest quarter, solar radiation, and anthropogenic disturbance. The future predictions under moderate and severe scenarios of global change indicated a drastic distribution area reduction, especially in the southern zone around Tarapacá and Antofagasta in Chile. These suggest a potential loss of unique genetic diversity and the need for conservation actions particularly focused at the edges of the H. markhami distribution.

Introduction

Seabird populations are threatened by the loss and degradation of breeding and non-breeding habitats, illegal killings, and climate change (Bairlein, 2016; Studds et al., 2017; Wilson et al., 2018; Xu et al., 2019), which have induces steep declines in abundance and distribution at broad scales. This problem is only exacerbated by inefficient conservation policies that do not consider the conservation of breeding sites during the development of investment projects (Syroechkovskiy Jr, 2006; De Boer et al., 2011), and major knowledge gaps in many species’ basic biology, especially in poor countries and for small-bodied species (Rodriguez et al., 2019). This is particularly troubling on one of the most endangered avian groups, the Procellariiformes (Class Aves) (Croxall et al., 2012), which are disproportionately threatened compared to Aves overall (Rodriguez et al., 2019). For the “storm-petrels” this is not only because of its small size, but also because of its cryptic breeding behavior (i.e., nocturnal colony visits, underground nesting, remote and inaccessible reproduction areas), high mobility that in some cases, prevent their study, management, and conservation (Brooke, 2018), and broad distribution that puts them under different regulations of many national and international jurisdictions and boundaries (Harrison et al., 2018). All these characteristics contribute to a high vulnerability of breeding storm-petrels to anthropogenic disturbances. Additionally, one of the main threats at storm-petrel breeding habitat are light pollution (e.g., Rodríguez et al., 2017). Especially for species with small, restricted breeding grounds near human populations, where light pollution induces high mortality of fledglings (e.g., Gineste et al., 2016).

The Markham’s Storm-Petrel (Hydrobates markhami) is one of the least known migratory seabirds in the world (Croxall et al., 2012; BirdLife International, 2019). However, artificial light from large cities near breeding areas are known to cause mortality from collision impacts and indirectly from predation by vultures on grounded individuals (Barros et al., 2019). This small pelagic species (21–23 cm) is found mainly in tropical waters of the Pacific Ocean, between 5°N and 29.9°S, and 71°W and 118.02°W (Murphy, 1936; Spear & Ainley, 2007; Howell & Zufelt, 2019). Hydrobates markhami is a colonial breeder, with five known dispersed colonies in the Sechura and Atacama Deserts, specifically in saline areas (i.e., salt flats) where they use fissures and cavities found under the surface on the salt flats for nesting, displaying strong philopatry to their natal colonies and nesting sites (Jahncke, 1993; Jahncke, 1994; Torres-Mura & Lemus, 2013; Schmitt, Barros & Norambuena, 2015; Barros et al., 2019). The northernmost colony is in Paracas, Peru, where the species breeds in small, dispersed colonies up to 5 km from the sea on the sloping ground (Jahncke, 1993; Jahncke, 1994). The other four colonies are in Chile, located in the Coastal Atacama Desert at up to 50 km inland (Fig. 1; Barros et al., 2019; Medrano et al., 2019). In Chile, the known colonies are (1) Arica, (2) Pampa de la Perdiz, (3) Salar Grande and (4) Salar Navidad (Medrano et al., 2019). Reproduction is asynchronous between colonies (Barros et al., 2019; Medrano et al., 2019). In the northern colonies in Paracas and Arica, most pairs lay eggs between April and August, and chicks hatch asynchronously from July to January with a peak between July and September (Jahncke, 1994; Barros et al., 2019; Medrano et al., 2019). While in the colonies of Pampa Perdiz, Salar Grande and Salar Navidad breeding pairs lay eggs between November and January, and chicks hatch between January and April (Barros et al., 2019; Medrano et al., 2019). The species population size is estimated at 2,305–4,362 breeding pairs in Peru (Jahncke, 1993; Jahncke, 1994) and 55,308–55,733 breeding pairs in Chile (Barros et al., 2019; Medrano et al., 2019). IUCN consider H. markhami as Near Threatened (NT; BirdLife International, 2019), but under Chilean classification, this species is categorized as Endangered (EN) due to decreasing population size and extended threats over the breeding colonies (Barros et al., 2019; Medrano et al., 2019).

Figure 1 Map of the phylogeographic structure of Hydrobates markhami.

Map of the phylogeographic structure of Hydrobates markhami showing: (A) the mtDNA ND1 Bayesian Inference (BI) and Maximum Likelihood (ML) phylogeny and the distribution of the three main clades (upper node values represent BI posterior probabilities and down nodes values represent ML bootstrap values). (B) Haplotype network and each locality; the scale represents the sample size for each locality. At the bottom is the map with the breeding sites used for the genetics analysis. Out-groups on phylogeny are not shown. Photograph of H. markhami: courtesy of Fernando Díaz Segovia.

Currently, the genetic data and species distribution modeling could provide important insights for conservation management of H. markhami, by examining the relationship between the environment and the species distribution (Humphries et al., 2012; Field et al., 2020). Additionally, it is important to include future scenarios in these combined analyses, given that for many threatened species it has been suggested that their distribution under climate change scenarios will experience important changes, due to translocation of habitat optima (Pecl et al., 2017; Beaumont et al., 2019; Field et al., 2020) or drastic reductions in habitat suitability (Cianfrani et al., 2018; Borges et al., 2019). Considering its complex conservation scenario and the singular breeding habitat geography between colonies and the strong philopatry in H. markhami (i.e., reproductive isolation and low migration) we expected to find narrow habitat distribution conditions, strong genetic geographic structure, and spatially differentiation related to human population activities (e.g., light pollution) and the climate global change. Under the climate change context, species with unique life history traits are expected to experience more changes in their distribution, which could even lead to loss of genetic diversity (Loarie et al., 2009; Pecl et al., 2017). The aims of this study were to: (1) evaluate the phylogeographic structure and genetic diversity distribution across most of H. markhami distributional breeding range; (2) evaluate the effect of climatic variables and anthropogenic impact proxies on the H. markhami geographic distribution using ecological niche modeling in a Bayesian framework; (3) quantitatively assess the vulnerability to climate change under two representative concentration paths (RCP) of greenhouse gas emissions, which correspond to future climate condition trajectories; and (4) evaluate the effect of migration capacity on the spatial distribution models.

Materials and methods

Phylogenetic analysis

Between November 12, 2018 and December 01, 2019, we collected blood samples from seven specimens of H. markhami from Pampa Chaca, Arica (18°70′S, 70°24′W) and 9 specimens from Salar Grande (21°01′55′S, 69°59′13′W) under permits from Servicio Agrícola Ganadero, SAG, of Chilean government (No 5022/2014 and 5742/2016). DNA was extracted from frozen samples following the protocol of Fetzner (1999) using the QIAGEN DNAeasy kit. We amplified the mitochondrial gene NADH dehydrogenase subunit I (ND1), via polymerase chain reaction (PCR). PCR was performed in a total volume of 25 µL containing 12.5 µL Thermo Scientific PCR Master Mix (0.05 U/µL Taq DNA polymerase, reaction buffer, 4 mM MgCl2, 0.4 mM of each dNTP), 0.2 µM of each primer and 20 ng of template DNA. Amplification was performed using the forward TMET-forward: 5′ACC-AAC-ATT-TTC-GGG-GTA-TGG-G 3′ and the reverse primer 16DR-reverse: 5′CTA-CGT-GAT-CTG-AGTT-CAG-ACC-GGA-G 3′ (Leaché & Reeder, 2002). The following thermal cycler settings were used to amplify all reactions: 5 min at 94 °C followed by 35 cycles of 94 °C for 30 s, 55 °C for 30 s, and 72 °C for 30 s, followed by a final extension of 72 °C for 5 min (Sausner et al., 2016). PCR products were sequenced in both directions through automatic sequencing using Macrogen’s ABI3730XL (Seoul, South Korea). Sequences were edited using Codon Code Aligner v. 3.0.3 (CodonCode Corporation, http://www.codoncode.com), and translated into amino acids to corroborate the absence of stop codons. We got a total of eight sequences with high quality for the analysis (MZ768852 to MZ768859; Table S1). Moreover, five ND1 sequences from Paracas breeding site in Cerro Lechuza, Peru (13°8′S, 76°8′W), and one sequence from Chaca in Arica, Chile (18°S, 70°W) were obtained from Sausner et al. (2016) in GenBank (Table S1). In order to avoid obtaining spurious outcomes resulting from the lost phylogenetic information due to substitution saturation, we tested whether the sequences used were useful for the phylogenetic analysis through Xia’s test (Xia et al., 2003) implemented in DAMBE v7 (Xia, 2018). Xia’s test is an entropy-based index that estimates a substitution saturation index (Iss) in relation to a critical substitution saturation index (Iss.c), by using a randomization process with confidence intervals (95%). The proportion of invariable sites for this analysis was determined in jModeltest 2 (Darriba et al., 2012). The sequences of H. markhami are available in GenBank according to the accession numbers provided in Table S1. As outgroups in the phylogenetic analyses, we used: H. melania, H. microsoma, H. tethys, H. hornbyi, H. leucorhoa and H. homochroa; representatives of Oceanitidae Oceanites gracilis and Fregetta grallaria; Diomedidae Thalassarche chrysostoma, Thalassarche melanophrys and Phoebastria nigripes; and Procellariidae Aphrodroma brevirostris (Table S1).

Data were analyzed as previously described in Norambuena et al. (2018), we used both Bayesian inference (BI) and maximum likelihood (ML) approaches for phylogenetic reconstruction. We conducted Bayesian analyses using BEAST v. 1.10.4 program (Drummond et al., 2012), using ‘Yule speciation process’ for the tree prior to consider the effect of divergent sequences on outgroups (Drummond et al., 2012). We identified the best-fit nucleotide substitution model using jModeltest 2 (Darriba et al., 2012), which indicated HKY+ Γ as the best-fit model for ND1 using BIC and AICc criterion. We ran all analyses for 100 million generations, and we sampled every 1,000 steps; the first 25% of the data was discarded as burn-in. The convergence of MCMC analysis was examined visually in Tracer v1.6 (Rambaut & Drummond, 2009).

ML analyses were conducted in RAxML v8 (Stamatakis, 2014) using the multiple inference strategy. We ran 1,000 independent inferences and 1,000 bootstrap replicates with the same nucleotide substitution model settings as for the Bayesian analysis. Support values from 1,000 bootstrap replicates were annotated on the tree with the highest likelihood. .

We also inferred a haplotype network as previously described in Norambuena et al. (2018) by using the “median joining network” algorithm in Network 4.610 (Bandelt, Forster & Röhl, 1999), which is based on the sum of weighted differences (i.e., Hamming distance) between sequences. Ambiguities within the network were solved according to the criteria of Crandall & Templeton (1993). Finally, for each geographic area (i.e., Paracas, Arica and Salar Grande) retrieved by phylogenetic analysis, we calculated in DnaSP v.5 (Librado & Rozas, 2009) the number of polymorphic sites (S), haplotype diversity (H) and nucleotide diversity (Π).

Occurrence data and climate variables

Hydrobates markhami occurrence data were obtained during 12 expeditions (see Barros et al., 2019 for details), literature from the Peruvian colonies (e.g., Jahncke, 1993; Jahncke, 1994; Torres-Mura & Lemus, 2013), and electronic databases (eBird, 2020). We obtained a total of 972 georeferenced records that were later reduced to 75 data cleaning (Data S1). All the final records correspond to confirmed and potential nests on breeding sites (Jahncke, 1993; Jahncke, 1994; Barros et al., 2019). To reduce spatial autocorrelation that usually results from sampling areas with a high density of locality points (clusters of points), we spatially filtered locality data to allow a minimum distance of 1 kilometer between any two points.

The climatic variables were obtained from Wordclim version 2.1 with 2.5 min spatial resolution (Fick & Hijmans, 2017). Additionally, we obtained environmental variables such as ultraviolet radiation (Beckmann et al., 2014), elevation, wind (Fick & Hijmans, 2017) and topographic roughness (Amatulli et al., 2018). As anthropogenic impact proxies, we used the databases of Human footprint (Venter et al., 2016), artificial lights (Falchi et al., 2016) and human population (CIESIN, 2016). We selected the variables that were used in the models through an exploratory analysis which resulted in strongly correlated variables to be eliminated. We used the variance inflation factor (VIF) to evaluate the collinearity among predictors, where VIF greater than 10 is a signal that the model has collinearity problems (Quinn & Keough, 2002). Our analyzes showed that four variables have a VIF < 3 and four variables have a VIF between 4 and 9, which are below the threshold (VIF < 10) (See Fig. S1 and Table S2). Finally, eight uncorrelated variables were used to perform the species distribution models (SDMs): Min Temperature of Coldest Month, Temperature Annual Range, Mean Temperature of Wettest Quarter, Mean Temperature of Driest Quarter, Human footprint, artificial lights, solar radiation, and wind (see Table S2).

Species distribution modelling (SDM)

The geographical distribution of H. markhami was modeled using an intrinsic conditional model (iCAR). We assume that the response variable Zi is a binary variable that represents the presence (1) or absence (0) of H. markhami. This approach explicitly considers spatial autocorrelation (Latimer et al., 2006) to adjust an ecological process where the presence/absence of the species is explained by the suitability of the habitat (Vieilledent et al., 2014), where:

Zi ∼ Bernoulli (πi)

logit (πi) = Xiβ+ ρj(i)

Xi = matrix of covariates, β = vector of the regression coefficients, ρ represents the random spatial effect of the observation i in cell j, and the logit link is used to model the relationship between πi, the covariates and spatial effect. Models were built using the package “hSDM” (Vieilledent, 2019) in the software R (R Core Team, 2019). Uninformative priors centered at zero with a fixed large variance of 100 were used for all parameters involved in both ecological and observation processes, while a uniform distribution was used for the variance of the spatial effects (Pennino et al., 2017). We chose the model which had the lowest Deviance Information Criterion (DIC) (Spiegelhalter et al., 2002), where lower values of DIC represent the best compromise between fit and estimated number of parameters.

Geographical projection to the future scenarios

To model the geographic distribution into the future, we selected two representative concentration paths (RCP) of greenhouse gases, which correspond to future climate conditions trajectories of greenhouse gases adopted by the IPCC (Stocker et al., 2013). RCPs span the range of the year 2100 radiative forcing, i.e., from 2.6 to 8.5 W/m2 (Van Vuuren et al., 2011). An RCP 2.6 was selected as the scenario for an extremely low forcing level and 8.5 as an extremely high baseline emission scenario. Three global climate models (GCMs); CCCMA, CSIRO and MIROC were evaluated for the year and 2080. The climatic projections were obtained from the portal http://www.ccafs-climate.org/ (Navarro-Racines et al., 2020). We explored 32 GCMs projections through the GCM compareR application (Fajardo et al., 2020). CCMA, CSIRO represent models where low precipitation and high temperature are represented, while MIROC is a more conservative model and closest to the study area’s average (e.g., Alarcón & Cavieres, 2015; Lazo-Cancino et al., 2020). In the future projections of the SDMs, we only use the variables min temperature of the coldest month, temperature annual range, mean temperature of the wettest quarter, and mean temperature of the driest quarter. The variables for Human footprint, artificial lights, radiation and wind were not included since there are no future predictions for these variables.

To quantify geographic distribution changes under future climate change scenarios, we compared the current model with future projected models. Each model was converted from a continuous output to a binary classification (presence/absence) using the threshold that maximizes the sum of sensitivity and specificity (max SSS) (Liu, White & Newell, 2013). Then the areas gained, lost, no occupancy, and no change in the future were estimated. We calculated the estimated areas in square kilometers using the South American Albers Equal Area Conic projection. We used SDMtoolbox module (Brown, 2014) implemented in ArcGIS 10.4.1 (Esri, Redlands, CA, USA) to calculate the areas of expansion range, contraction range, and the distribution without change between present and future models.

Migration constraints

Since the SDMs are static in nature and do not consider the species dispersal ability, biotic interactions, or population dynamics (Zurell et al., 2009; Zurell et al., 2016; Engler, Hordijk & Guisan, 2012), these do not allow predicting the effect of climate change on the distribution of species in a realistic way. Therefore, in our models, we consider limitations to the dispersion of H. markhami in the projections, through the “Migclim” approximation (Engler et al., 2009; Engler, Hordijk & Guisan, 2012). This method explicitly includes the dispersal of a species, potential propagule production, geographic barriers, short-distance dispersal capacity (SDD) and probability for long-distance dispersal (LDD) (Engler, Hordijk & Guisan, 2012). We performed the analyzes using the habitat suitability models predicted by iCAR for the years 2030, 2050 and 2080. Because not all the parameters required by the algorithm are known, we opted to consider a model with barriers to dispersion, using three thresholds (300, 500 and 700), where habitat suitability scales from 0 to 1000, and the values below the threshold are considered absences and above the threshold they are considered presences. These thresholds allow the classification of suitable or unsuitable habitat, where cells with habitat suitability ≥ threshold are considered as suitable, values <threshold unsuitable. We do not consider long-distance dispersion since H. markhami is highly philopatric (see results). To set the spatial barriers for future dispersion we consider the Human Footprint as a “strong” barrier, given the sensitivity of this species to human activity. The analyzes were performed for the three global climate models (GCMs); CCCMA, CSIRO and MIROC. We compare the results of each Migclim run selecting the best and worst simulated scenario, as follow: for the best scenario, we consider one with the highest number of occupied cells, the smallest number of absent cells at the end of the dispersion process, and the largest number of cells that could be used in the case of ”unlimited dispersal” and ”non-dispersal”; and the opposite conditions was considered as the worst scenario. The analyzes were performed in the MigClim 1.6 package (Engler, Hordijk & Guisan, 2012). All simulations for each GCMs and RCP are detailed in Table S3.

Results

Genetic population structure

Sequences of 955 bp in length for the ND1 locus were obtained and the result of Xia’s test suggests low saturation, as the critical index of substitution saturation value (Iss.c = 0.819) were significantly higher than the observed index of substitution saturation values (Iss = 0.514; p < 0.0001), therefore, the sequences were deemed suitable for performing phylogenetic analyses. Four haplotypes were identified defined by 17 polymorphic sites. The ML and BI trees based on ND1 sequences showed identical topologies (Fig. 1A). Both trees inferred that the H. markhami is monophyletic and composed of three well-supported clades (posterior probability pp of 0.9–1.0 and ML bootstrap support of 100). The three clades are geographically structured, with one clade represented by Paracas individuals, the other by Arica, and the third by Salar Grande individuals (Fig. 1A). The only geographic incongruence was in the Paracas clade where a sample from Arica was the sister of all the Paracas individuals that are monophyletic with a 0.9 of pp (Fig. 1A).

The ND1 haplotype network revealed the same three major clades recovered from the BI and ML phylogenies (Fig. 1B). Overall haplotype diversity for the ND1 gene was 0.747 ± 0.004 and overall nucleotide diversity was 0.028. The clade of Paracas had the highest haplotype diversity and the clade of Arica had the highest nucleotide diversity, while the clade of Salar Grande had the lowest values and was represented by one exclusive haplotype.

Present and future geographic distribution

The H. markhami distribution was mainly driven by min temperature of the coldest month (Bio 6), temperature annual range (Bio 7), mean temperature of the wettest quarter (Bio 8), mean temperature of the driest quarter (Bio 9), Human footprint (HFP), and radiation (Table 1). Positive relationships were found between Bio 8, Bio 9, HFP, radiation and the occurrence of H. markhami, and negative relationship with the variables Bio 6 and Bio 7 (Table 1, Fig. S1). The highest median posterior probability of the presence of H. markhami occurs around the region of Paracas (Peru), and in a continuous area from the southern coast of Arequipa region in Peru to Antofagasta region in Chile (Fig. 2A). The spatial effect, that is, a model without considering the environmental predictors, showed to be strong for almost the entire modeled distribution range of the species (Fig. 2B). By 2080, the distribution predicted considering less severe scenarios (i.e., RCP 2.6) and severe (RCP 8.5) showed that expansion areas geographic range were greater than the areas of contraction geographic range (Figs. 3 and 4) (See Table S3 and Fig. S2). The MIROC model (RCP 2.6) was the only case where there would be a greater contraction of the geographic range predicted in the future (Table S3). In all models and Representative Concentration Pathway, a high no-occupancy geographic area is predicted, that is, areas currently not occupied by the species and that are not expected to be occupied in the future. Similarly, a significant area of no-change is predicted, that is, areas currently occupied by the species and expected to remain occupied in the future (See Table S3 and Figs. S2–S5).

Table 1 Summary of the fixed effects posterior distribution for the best model of the H. markhami.

Mean, standard deviation (SD), and a 95% credible interval containing 95% of the probability under the posterior distribution (Q0.025-Q0.975).

	Mean	SD	Q0.025	Q0.975	
Intercept	−17.732	1.685	−20.571	−14.448	
Bio 6	−14.658	4.128	−25.068	−9.147	
Bio 7	−5.682	2.525	−10.823	−1.414	
Bio 8	5.348	1.902	1.575	9.054	
Bio 9	14.750	2.407	10.725	19.674	
HFP	1.251	0.539	0.189	2.331	
Light	0.524	0.758	−1.023	2.061	
Radiation	3.141	2.028	0.179	6.65	
Wind	1.595	1.658	−1.241	4.207	
Vrho	75.605	3.492	71.218	84.312	
Deviance	45.966	9.403	29.630	65.917	
Notes.

Bio 6 min temperature of coldest month

Bio 7 temperature Annual Range

Bio 8 mean temperature of wettest quarter

Bio 9 mean temperature of driest quarter

HFP human footprint

Figure 2 (A) Median of the posterior probability of the presence of the Hydrobates markhami, (B) spatial effect (the spatial component represents the intrinsic spatial variability of the data without variables).

Figure 3 Habitat suitability maps for future climatic conditions predicted for 2080 under a RCP 2.6 (benign scenario).

(A) Map of habitat suitability under GCMs CCCMA, (B) map of habitat suitability under GCMs CSIRO, and (C) map of habitat suitability under GCMs MIROC.

Figure 4 Habitat suitability maps for future climatic conditions predicted for 2080 under a RCP 8.5 (hard stage).

(A) Map of habitat suitability under GCMs CCCMA, (B) map of habitat suitability under GCMs CSIRO, and (C) map of habitat suitability under GCMs MIROC.

The habitat changes predicted by the simulations indicate two highly contrasting simulations (Table 2). The first one indicating a low impact on the distribution of the species in a future scenario for the GCMs model MIROC (RCP 2.6) with a high number of pixels that would be colonized at the beginning and end of the simulation, though, there are also numerous areas to the south of their known distribution that would not be colonized (pink pixels) (Fig. 5A, Table 2); The second, being the worst case scenario for H. markhami (Fig. 5B, Table 2), where the CCMA model (RCP 8.5) indicated a low number of occupied and colonized pixels at the end of the simulation, reducing the number of areas that are suitable in the present and that would also be suitable in the future (red pixels). The remaining simulations by GCMs and RCP support previous results and are shown in Figs. S2–S4 of the supplementary materials.

Table 2 Expected change in habitat (number of pixels) by simulation.

GCMs	RCP	Thresh-old	No dispersal
count	Unlimited
dispersal
count	Occupied
count	Absent
count	Total
colonized	Total
decolonized	
	8.5	300	44	966	656	1234864	675	64	
	2.6	300	44	909	813	1234707	878	110	
CCMA	8.5	500	42	421	390	1235130	405	60	
	2.6	500	41	434	414	1235106	433	64	
	8.5	700	38	197	195	1235325	190	40	
	2.6	700	37	215	209	1235311	201	37	
	8.5	300	44	1271	661	1234859	699	83	
	2.6	300	44	1073	851	1234669	876	70	
CSIRO	8.5	500	42	691	490	1235030	480	35	
	2.6	500	42	371	352	1235168	385	78	
	8.5	700	37	351	292	1235228	269	22	
	2.6	700	35	149	145	1235375	169	69	
	8.5	300	43	1140	664	1234856	700	81	
MIROC	2.6	300	44	1227	916	1234604	923	52	
	8.5	500	42	580	431	1235089	487	101	
	2.6	500	42	587	471	1235049	439	13	
	8.5	700	37	228	196	1235324	255	104	
	2.6	700	36	276	202	1235318	178	21	
Notes.

GCMs general circulation models

RCP Representative Concentration Pathway

Threshold value to change a continuous prediction to binary

No Dispersal number of cells that would be occupied in the case of the No Dispersal scenario

Unlimited Dispersal Number of cells that would be occupied in the case of the Unlimited Dispersal scenario

Occupied number of cells that are in an occupied state at the end of the given dispersal step

Absent Number of cells that are in an unoccupied state at the end of the given dispersal step

Total Colonized Number of cells that turned into an occupied state

Total Decolonized Number of cells that turned into an unoccupied state during the given dispersal step

Figure 5 MIGCLIM output map Dispersal restricted future distribution of Hydrobates markhami, under two RCP.

(A) Simulation for model MIROC (RCP 2.6). (B) Simulation for model CCCMA (RCP 8.5).

Discussion

Genetic population structure

The phylogeny and haplotype network supported three main lineages within H. markhami, showing a clear geographic structure associated to breeding areas in Paracas, Arica and Salar Grande. The shared haplotype between Paracas and Arica suggests some degree of connectivity between both areas (gene flow), however, we cannot discard that could be due to incomplete lineage sorting. This result is coherent with the fact that the northern colonies (Paracas and Arica) share breeding phenology, with most pairs laying eggs between April and August, and chicks hatching from July to January (Jahncke, 1994; Barros et al., 2019; Medrano et al., 2019). On the other hand, the differentiated haplotype from Salar Grande shares breeding phenology with Pampa Perdiz and Salar Navidad, with pairs laying eggs between November and January, and chicks hatching between January and April (Barros et al., 2019; Medrano et al., 2019). This fact supports the importance of breeding phenologies as a key factor in explaining microevolutionary processes in the Hydrobates genus. For example, for the H. castro species complex, the occurrence of two different phenologies (hot and cold season) has been described as a relevant mechanism for sympatric speciation (Monteiro and Furnes, 1998; Friesen et al., 2007). This strong degree of geographic structure has been documented previously in Peruvian diving-petrel and two Patagonian shag species that breed in colonies along the coasts of Peru, Chile and Argentina (Calderón et al., 2014; Cristofari et al., 2019). However, to test the relevance of the gene flow hypothesis requires additional samples improve our preliminary result about geographic structured pattern and active dispersion between Paracas and Arica.

The biological association between breeding phenologies in H. markhami and saline areas produce strong philopatry to their natal colonies and nesting sites (Jahncke, 1993; Jahncke, 1994; Torres-Mura & Lemus, 2013; Schmitt, Barros & Norambuena, 2015; Barros et al., 2019). A large number of studies have now documented that all Hydrobates species of South America use salt flats/saltpetre deposits in the coastal deserts of Sechura and Atacama to nest (Jahncke, 1993; Jahncke, 1994; Bernal, Simeone & Flores, 2006; Ayala & Sanchez-Scaglioni, 2007; Torres-Mura & Lemus, 2013; Barros et al., 2019; Medrano et al., 2019), and even some Oceanites storm-petrels (Oceanitidae family) have been found using the same areas for nesting (Barros et al., 2020).

Geographic distribution

Under conservative and more severe climate change scenarios, our models suggest moderate reductions and strong reduction of the distribution of H. markhami, respectively, which agree with the idea that species with specialist reproductive habitat (e.g., breeding phenology associated with saline areas) are especially sensitive to the effects of rapid climate-change (Loarie et al., 2009), because of the constraints imposed by this specific requirement for breeding sites (i.e., niche conservatism). To date, after multiple expeditions searching for breeding sites, they have only found H. markhami breeding in this specific environment (Barros et al., 2019; Medrano et al., 2019). But considering that some petrels are able to use cavities in other substrates, such as soft soil and man-made burrows (Podolsky & Kress, 1989; Bolton et al., 2004), we do not rule out the capability of nest substrate plasticity.

The most impacted area of H. markhami distribution will be its current southern edge between the Tarapacá and Antofagasta regions in Chile. Moreover, this is also the most affected area by light contamination (see Barros et al., 2019), considering that 11.41% (2.269 MW) of the electrical power of Chile is generated in this area and large cities dependent on mining-related economy (e.g., Iquique and Antofagasta) continue to grow. The future reduction in the distribution of H. markhami may be even more severe than suggested by our models, given that we were not able to include a prediction of human footprint in the models.

According to our results, the predicted future habitat range of H. markhami is likely to be negatively affected by future climate change and will concentrate on the central portion of its present distribution (i.e., around the Arica area). The first response of species to climate change under GCMs MIROC RCP 2.6 scenario would be the colonization of areas around present distribution, but under CCMA model (RCP 8.5) this colonization considerably decreased. In both scenarios the southern distribution would not be colonized and suitable in the future (MigClim simulation, Fig. 5). The MigClim simulations also suggest the loss of some areas in Southern Peru and in the south of Arica colony. Unlike species with greater mobility or fewer restrictions on reproductive habitat, that shift their distributions moving poleward and to higher elevations (Chen et al., 2011; Pecl et al., 2017), the limited distribution of salt flats/saltpetre deposits (Sáez et al., 2012)—key habitat for H. markhami nesting- will affect the responses of this species to climate change. This, added to the small population of H. markhami (Barros et al., 2019), would constrain its distribution range change by limiting its colonizing capability, and thus increasing its extinction risk.

Most research on the response of seabird to climate change has been studies considering at-sea distribution (e.g., Wolf et al., 2010; Humphries et al., 2012). However, for seabirds such as H. markhami, the individuals at the breeding colonies could be affected by the warming of air temperature, that in severe cases could cause mortality due to overheating and physiological stress (Sydeman, Thompson & Kitaysky, 2012). These last conditions, related to an increase of temperature, could be particularly important in the breeding habitat of H. markhami in the Sechura and Atacama Deserts. In fact, it is expected that seabird species will respond differentially to climate change according to many different factors, including life history characteristics, diet, range, and abundance (Furness & Tasker, 2000; Sydeman, Thompson & Kitaysky, 2012). So, while some seabirds may fare well in warming oceans, others may become locally, regionally, or perhaps even globally extinct (e.g., Kitaysky & Golubova, 2000; Jenouvrier et al., 2009; Wolf et al., 2010; Lewison et al., 2012).

Conclusions

Overall, our results of H. markhami can be useful for the design of conservation policies, considering that the planning of protected areas and management should be focused on areas with higher or unique genetic diversity (Midgley et al., 2003; Ayebare et al., 2018). In H. markhami, the extinction of any local population could mean a loss of unique genetic diversity. The southern portion of the H. markhami distribution (Tarapacá and Antofagasta) are the most vulnerable areas according to our results and do not have any type of legal protection today. The only breeding area partially protected is Paracas in Perú (Jahncke, 1993; Jahncke, 1994). This provides a complex conservation scenario for this species, especially considering the future consequences of climate change. Finally, considering the complex conservation scenario, singular breeding habits, its narrow habitat distribution conditions, preliminary evidence of genetic geographic structure, and spatial differentiation related to human population activities (e.g., light pollution and climate global change); we highlight the urgent need for increased cooperation and governance between the Peruvian and Chilean wildlife technical units, and the protection of their breeding sites in the center and south of their distribution, given that the local extinctions occur closer to the border or core range depending on local and regional environmental factors intermingled with human impacts (Cowlishaw, Pettifor & Isaac, 2009; Boakes et al., 2018).

Supplemental Information

Supplemental Information 1 Supplemental figures

Click here for additional data file.

Supplemental Information 2 Taxon sample list, tissue number, country/locality and GenBank accession number

Click here for additional data file.

Supplemental Information 3 Pearson correlation matrix of environmental variables used in ecological niche modelling

Click here for additional data file.

Supplemental Information 4 Predicted contraction, expansion, areas of no change and no occupancy (Km2) for the distribution of Hydrobates markhami. RCP= Representative Concentration Pathway

Click here for additional data file.

Supplemental Information 5 Occurrence dataset of Hydrobates markhami.

We obtained a total of 972 georeferenced records that were later reduced to 75 data cleaning. All the final records correspond to confirmed and potential nests on breeding sites (see Jahncke, 1993; Jahncke, 1994; Barros et al., 2019; eBird, 2020).

Click here for additional data file.

We thank Fernando Medrano for his comments on drafts of the manuscript. We thank the ‘ROC’ volunteers that assisted the research and conservation project of Atacama storm-petrels. We thank an anonymous reviewer and Marcelo Rivadeneira for revisions to the manuscript.

Additional Information and Declarations

Competing Interests

Author Contributions

Animal Ethics

DNA Deposition

Data Avialability

The authors declare there are no competing interests.

Heraldo V. Norambuena conceived and designed the experiments, performed the experiments, analyzed the data, prepared figures and/or tables, authored or reviewed drafts of the paper, field work, and approved the final draft.

Reinaldo Rivera conceived and designed the experiments, performed the experiments, analyzed the data, prepared figures and/or tables, authored or reviewed drafts of the paper, and approved the final draft.

Rodrigo Barros, Rodrigo Silva and Ronny Peredo conceived and designed the experiments, authored or reviewed drafts of the paper, field work, and approved the final draft.

Cristián E. Hernández conceived and designed the experiments, performed the experiments, authored or reviewed drafts of the paper, and approved the final draft.

The following information was supplied relating to ethical approvals (i.e., approving body and any reference numbers):

Servicio Agrícola y Ganadero approved the study (N°5022/2014 and 5742/2016).

The following information was supplied regarding the deposition of DNA sequences:

The ND1 sequences from Hydrobates markhami are available at GenBank: MZ768852 to MZ768859.

The following information was supplied regarding data availability:

DNA sequence accession numbers are available in Table S1.

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
