# Peer review of "Living on the edge: genetic structure and geographic distribution in the threatened Markham’s Storm-Petrel (Hydrobates markhami)"

_PeerJ, doi:10.7717/peerj.12669_

## Round 0.1 · original submission · Major Revisions

As you can see, the two reviewers have suggested an extensive set of corrections. After revising, I will want to send your revision back to the reviewers for further comments.

Reviewer 1 ·

Basic reporting

As I describe in the General Comments to the authors, there are a number of issues with regard to the writing. Some of these issues may stem from English being a second language for the authors, but some are also matters of clarity and depth of description. The abstract and the discussion sections have the most issues, though there are a number throughout the manuscript. I think the writing needs a good polishing before it can become suitable for publication.

Also, some citations are missing from the reference list. I suggest the authors check this.

Otherwise, the background provided is sufficient and relevant, the data collected and the analyses conducted are relevant to address the question specified and provide insights that will be useful to conservation of a threatened species. The structure of the manuscript is professional, as are the tables and figures, though the latter require some revisions to allow them to stand on their own and to make them more easy to see/interpret.

Experimental design

The research question is well defined and relevant. It fills a knowledge gap and will be important for conservation of a threatened species.

However, I believe the methods are not currently described in sufficient detail for them to be independently replicated. Please see the General comments to the authors for more information, but in general there are a number of missing details (e.g. PCR recipe, how literature records were filtered, etc).

I also have some concerns about the data analyses carried out (again pleae see the general comments to the authors for more information). None of these are fatal issues, but I suggest some more thoughtful description and potential reanalysis could show that they are robust and valid.

Validity of the findings

Overall, the results found are novel and provide useful information. I had some concerns about the robustness of the specific data analyses implemented (described above) but the general approach is sound.

Some assumptions are made (e.g. that storm petrels have strong fidelity to their nesting habitat), which are not supported with the literature. Also, some conclusions are made to strongly and the data/results presented in the manuscript are not sufficient to support them. They need to be more clearly identified as speculation, or removed/rephrased.

Additional comments

In this paper, the authors describe genetic and geographic distribution model analysis for a threatened seabird, Markham’s storm-petrel.

Overall, I think the data/results presented in this paper are important for conservation on this seabird and would be worth publishing. However, in its current form the manuscript lacks clarity on many aspects, has some issues regarding data analysis, and makes some (strong) conclusions without sufficient evidence/data. I think these issues would need to be addressed before the manuscript could be suitable for publication, and I have detailed them below, along with some more minor comments. I hope the authors find these comments helpful in revising their paper.

Writing: I can appreciate that English may not be the first language of the authors, however there are many places throughout the manuscript where the writing needs to be revisited, either for grammar or for clarity. The abstract and the discussion in general have the greatest number of these issues. However, there are other instances scattered throughout the manuscript. Some examples (but NOT an exhaustive list):
• Line 44 – What do you mean by inefficient conservation policies? Can you be more specific?
• Line 60 – Artificial light can’t cause predation, so something seems missing here.
• Line 65 – It would be best at first use to describe what you mean by saline areas in a desert
• Line 143 – “Bootstrap support values were passed to the tree with the highest likelihood among the 1,000 independent tree inferences.” Please clarify as this currently doesn’t make sense. Perhaps you mean something like ‘Support values from 1000 bootstrap replicates were annotated on the tree with the highest likelihood’?
• Line 262-266 – This description of the results is too vague. Please be more precise and descriptive, rather than referring just to the figures.
• Line 331-332 - ???

There were several issues with data analysis that I think need to be addressed prior to publication. I have listed them below by section.
• Phylogenetic analyses: Coalescent constant size was used as the tree prior for the BEAST analysis, and the justification provided is that these are population level analyses. That’s true, but if you include the outgroups (e.g. Albatrosses) there is quite a deep set of sequences used in the analyses, and presumably these are only a single sequence per taxon. I worry that this might have been an invalid prior (perhaps this is why you needed to run for 100 million generations with such a small data set?). I suggest repeating the analysis using a different tree prior (Birth-death or Yule) to see how it might influence your results.
• Occurrence: The methods here are not clear enough to allow the work to be reproduced. For example, a total of 972 georeferenced records were initially identified and then these were reduced to 75 (line 160) but the criteria for removing all these records was not described. Further, if all the records correspond to breeding sites that were already known, what was the value of doing the literature search in the first place? Where did the anthropogenic disturbance data come from, specifically? Can it be found in the papers listed (e.g. supplemental data?) or from some databases? If the latter, then give the name of the database and how to access it.
• Occurrence: The variance inflation factor was used to remove variables that were related to each other, and the threshold set was 10. These thresholds have been found to have some issues (e.g. see A caution regarding rules of thumb for variance inflation factors RM O'brien - Quality & quantity, 2007). I suggest defending your choice of threshold more thoroughly, and also acknowledging the limitation of this kind of analysis in the text. Further the Quinn and Keough 2002 citation is not in the reference list – I presume this is the textbook?
• Geographic projection for future scenarios: Please explain/defend why you chose those three climate models (CCCMA, CSIRO, MIROC). How do they differ?
• Migration constraints: Line 219 – please provide more information about the three thresholds selected (300, 500, and 700). Are these km? That will help put the dispersal analysis into a better context for the reader.

There are also a number of instances where the interpretation of the results is too broad or too strong for the data presented in the paper. Here are three examples from the genetics section:
• Line 282 – I think you need to add the word potentially here. The observed pattern with the haplotypes could be do to gene flow, but it could also be due to incomplete lineage sorting of ancestral variation.
• Line 297 -299 – I think there is not enough data presented in this paper to support this statement. Figure 1 shows that the north lineage is sister to a clade of more southern lineages. There is no dating or ancestral state reconstruction evidence to support this statement.
• Line 307 – 312 – There is not enough evidence in this paper to support this statement. Wallace et al. has the proper sampling to address this, whereas this paper does not. Further there is no evidence to date the splits among the Hydrobates storm petrels to say whether divergence occurred before/after/at the same time as the origin of the Sechura and Atacama deserts.

The authors seem to assume that these storm petrels can only breed in these desert areas. I think some discussion of the evidence for this assumption is warranted, as it drives the interpretation of the future predictions of suitable habitat. I know some burrow nesting petrel species (e.g. Pterodroma petrels and Shearwaters) will burrow in rocky outcrops AND in soft soil, plus they have bred in man-made burrows for conservation purposes. This suggests some plasticity. Also, the birds must have some level of dispersal, otherwise there would not be multiple colonies of them. Finally, double check the use of the word phenology (e.g. see line 317). I think it is usually used with regard to timing rather than habitat.

Data accessibility: I noted that there were Genbank accession numbers provided for sequences obtained from the database, but there were no accession numbers provided for the new sequences described here (though I see a BankIt submission file to Genbank). The accession numbers should be provided before publication. Also, there is reference to material presented in an Appendix (e.g. lines 170, 174), but I could not find any file/document that I interpreted to be the Appendix. Please either clarify or add it during the submission process.

Figures: The figures for the habitat suitability and future projections (Figures 3 -5 in the main text, and other supplemental figures) are very difficult to see because the range of the storm petrel is rather narrow and long. Is there anyway to make this easier? For example, if there is a large portion of unsuitable habitat in the middle or at the bottom, maybe you could just say that’s unsuitable in the text/figure legend and then ‘zoom in’ onto the areas where there are changes or differences between the models. Perhaps the Zoomed in figures could be presented in the main text and the full figures (as they are now) could be provided in the supplement.

Supplemental Figures S2, S4 – Why are some combinations of parameters not present here? E.g. RCP2.6 with threshold 300 in Figure S4 and RCP 8.5 with threshold 700 in Figure S2.



Minor comments

Title – This title suggests that the main emphasis of the paper will be describing a complex conservation scenario. I took this to mean, e.g. translocations, etc. The paper doesn’t really discuss the conservation scenario necessary, so I would revise. Perhaps something like “Living on the edge: genetic structure and geographic distribution in the threatened seabird Markham’s Storm-Petrel”

Line 72 – Refer to figure 1 to help the reader get a sense of distance between breeding areas

Line 75 – Are you referring to chick hatching here, or chick rearing/fledging? I’m surprised that the dates are so wide. Do they differ by colony or does each colony really exhibit the full range of dates?

Line 115 – Please provide details of the PCR reaction (e.g. the amount of buffer, type of taq used, etc). If it’s described in the same paper as the primer sequences, then please state that.

Line 123 – Is this previously published Arica sequence from the same breeding colony as the others?

Line 128 – Please paraphrase this description of the saturation test better. It seems to be very similar to text from another publication.

Line 132 – I think it would be better if you mentioned the outgroup sequences that you got from Genbank here. I found them later on, but think it makes more sense to describe them at the end of this paragraph. Also, please list all the species used as outgroups (‘representatives of Oceantitidae, Diomedidae and Procellariidae’ is too vague). Double check spelling of Procellariidae – it should have two i’s.

Line 137 – What criterion did you use to select the substitution model? AIC? BIC? AICc?

Line 153 – You list some parameters here that were not described later in the results (e.g. Fst, which the sample size is too small for anyway). Please double check what you list here vs. the results you present.

Line 203 – I find it a bit confusing that you give the future climate variables different names (bio 6, bio 8, etc) because then I have to look up what they are. I suggest just using a brief description of the variable (e.g. min temp for bio 6).

Line 239 – Were the four haplotypes described here what you found with your new sequences? It would be helpful to clarify. Also, you should describe the whole data set, including the genbank sequences. How many haplotypes there were in total, etc.

Line 244 – Is the strange Arica sequence from Genbank or from your own field work? Could there be an misidentification error/typo for that collection locality of that individual?

References: Please check that all the citations are in the reference list. I found a couple that were missing.

Table 1 and Table 2 – Say what analysis this table of results is for. The table should be able to stand on their own and it’s not clear from the caption what these data are about.

Figure 1 – The branch supports are too hard to read, please make the text larger. The phylogeny is too hard to see, I suggest making the lines darker. What is the orange colour in this figure?

·

Basic reporting

This article is aimed at describing the phylogeographic structure and the role of environmental variables affecting the geographic distribution of an endangered (Markham’s storm-petrel), making inferences about future distribution expected under different climate change scenarios/models. My general assessment is that the manuscript contains interesting and novel results for an endangered and little studied species, but that in an attempt of being succinct felt short in more detailed explanations for many of the methods used.

Importantly, figures are blurred, and the legend for Fig 5 (which seem to be key for further interpretations) is barely readable. In addition, it would be of great benefit providing insets showing a zoom to current nesting areas. The scale of the maps (Fig 2-5) is also too exaggerated; there is need of showing half of South America, specially if the study is focused on the three major nesting areas, and a more restricted map such as the one presented in Fig. 1 would do a better job.

Experimental design

One of my main concerns is the little connection made between the phylogeographic analyses and the SDM. These analyses (along the migration analysis) should be closely tied among each other. For instance, given the existence of these three large spatial clusters with clear differences in phenological breeding, would not be wiser to carry out a SDM for each subpopulation by separate? I guess you sorted that out using the MigClim approach, but that should be explicit in the text.

The migration analyses was quite novel (I did not even that existed!), but I would like to see a more in-depth explanation of MigClim simulation (I do not know what columns in Table 2 mean). Please note that the migration is not even one of the declared aims of the study, and appears out of the sudden in methods. Also, the results of the simulation are not even discussed. The MigClim simulation should be part of aims of the study, along with one or several hypotheses. Please also note that given the existence of these three large spatial clusters, it would be possible to estimate migration among subpopulations (with software such as MIGRATE) and complement the MigClim simulation. I am no expert of phylogeographic analyses, authors may correct me if this migration analysis is not suitable with the used markers.

Regarding the SDM, I applaud the use of good practices such as a curatorial procedure to discard bad spatial occurrences, but please indicate if that was carried out manually of using some automated procedures (e.g., Zizka et al 2019, Methods Ecol Evol. 2019;10:744–751). I agree with other decisions made by the authors, such as controlling by predictor collinearity, and spatial autocorrelation using of the iCAR model. However, these methods/decision are introduced without any context. There are legions of papers addressing these topics, and it would be good to re-direct readers to these original sources motivating your methodological decisions. Keep in mind that many authors use SDM as a black box, e.g., using Maxent by default with little regard of all caveats and assumptions needed.

On a minor note, a recent article discuss the importance of adopting standardized practices reporting the output of SDMs (Zurell, D., J. Franklin, C. König, P. J. Bouchet, C. F. Dormann, J. Elith, G. Fandos, X. Feng, G. Guillera‐Arroita, and A. Guisan. 2020. A standard protocol for reporting species distribution models. Ecography 43(9):1261-1277). It would be good for the article to adopt such practice.

Validity of the findings

I am puzzled by author's interpretation that "...conservative and more severe climate change scenarios suggest reduction and strong reduction of the distribution of H. markhami..." (lines 315-316). I simply fail to see a reduction in the geographic areas of distribution (comparing the blurred and out scaled Figures 2-4, please see my recommendation for improving figures above). Judging by the figures, I see that all three main nesting areas will remain habitable under all models and scenarios. Then how a strong reduction in distribution is concluded? I guess that the MigClim simulation is leading that interpretation, but since it is barely explained in the results (and not in the discussion) my question remains. If this is the case, and the whole case for the 'complex conservation scenario' is rooted on the MigClim simulation, then this approach should be properly highlighted in the introduction.

The second paragraph of Discussion (lines 300-312) is a bit disconnected from the actual results of the study. However, these ideas may be better connected by further phylogeographic analyses. For instance, a Bayesian skyline plot may inform about the temporal trends in the effective population size, which may be connected with Pleistocene/Holocene paleoclimatic reconstructions of the Atacama Desert.

---

## Round 0.2 · Minor Revisions

The remaining request of this one reviewer is minor indeed. So, please address it quickly and return the manuscript so that I can accept it.

·

Basic reporting

The revised version of this manuscript addresses most of my previous concerns. Authors have done a great job filling the gaps of the previous version, without a cost for the manuscript total length. Figure quality and resolution have also been improved.

A minor note
In the legend Figure 5the ‘pixel that have never been occupied (…)’ should be in gray symbol, not white (which is the sea in your map).

Experimental design

The new version of the manuscript addresses three major concerns I had before: 1) better connection between phylogeographic and SDMs analyses, 2) better explanation of curatorial procedures used to ensure data quality for SDMs, 3) a better contextualization and explanation of the MigClim simulation, and 4) an expanded explanation and justification for the use of the iCAR model.
My only remaining concern is that the raw database of geo-occurrences was not made available. Or is it already available in Barros et al. 2019? As far as I remember, having all the raw databases available is mandatory before final acceptance in the journal.

Validity of the findings

Now it is much clearer that the conclusions of the study are driven by the MigClim model. In fact, the MigClim is now more integrated to the rest of text.

---

## Round 0.3 · accepted · Accept

Thank you for making the minor decisions quickly. Such an interesting bird — I saw it on a pelagic trip to Lima a few years ago. Thanks for choosing PeerJ